# Tapentadol Immediate Release (IR) versus Morphine Hydrochloride for Postoperative Analgesia of Patients Undergoing Total Abdominal Hysterectomy—A Prospective Cohort Study

**DOI:** 10.3390/medicina59101800

**Published:** 2023-10-10

**Authors:** Sanja Starčević, Dragana Radovanović, Svetlana Škorić-Jokić, Milica Bojanić-Popovicki, Suzana El Farra, Nora Mihalek, Danica Golijanin, Tihomir Dugandžija, Ana Tomas Petrović

**Affiliations:** 1Faculty of Medicine, University of Novi Sad, 21000 Novi Sad, Serbia; dragana.radovanovic@mf.uns.ac.rs (D.R.); danica.golijanin@mf.uns.ac.rs (D.G.); tihomir.dugandzija@mf.uns.ac.rs (T.D.); ana.tomas@mf.uns.ac.rs (A.T.P.); 2Oncology Institute of Vojvodina, 21204 Sremska Kamenica, Serbia

**Keywords:** acute postoperative pain, tapentadol, abdominal hysterectomy, analgesia, morphine

## Abstract

*Background and Objectives*: This study aimed to examine the efficacy of tapentadol immediate release (IR) and morphine hydrochloride in the treatment of acute postoperative pain after total abdominal hysterectomy, as well as to examine the frequency of opioid-related side effects in observed patients. *Materials and Methods*: The prospective observational study was conducted over five months, and it included a total number of 100 patients. The two cohorts had different types of postoperative analgesia, and the effects were observed for 24 h postoperatively, by following the pain scores on NRS (Numerical Pain Scale), contentment with analgesia, and opioid-related side effects. *Results:* Statistical significance was found when assessing pain 24 h after surgery while coughing, where patients in the tapentadol IR group had significantly higher mean pain scores (*p* < 0.01). The subjective feeling of satisfaction with postoperative analgesia was statistically significant in the tapentadol IR group (*p* = 0.005). Vertigo appeared significantly more in patients from the morphine group (*p* = 0.03). *Conclusions*: Tapentadol IR (immediate release) and morphine hydrochloride are both effective analgesics used in the first 24 h after total transabdominal hysterectomy. Overall satisfaction of patients with analgesia was good. The frequency of side effects was higher in the morphine group, with statistical significance regarding the vertigo.

## 1. Introduction

Every surgical procedure causes trauma and stress to the organism, and it is followed by different pain intensities that a patient experiences. Pain is caused by tissue damage and the placing of the different prostheses and drains, and it is worsened by surgical complications. The pain intensity that a person experiences depends on preexisting comorbidities, among other things. In the normal postoperative course, the pain intensity decreases with wound healing, and it is considered that this process takes up to 3 months. If the pain is still present after that period, it becomes chronic pain and it demands a different treatment approach [1]. Studies have shown that inadequately treated acute postoperative pain can cause many problems, such as prolonged recovery after surgery and increased risk of developing venous thrombosis, pulmonary thromboembolism, postoperative ileus, pneumonia, insomnia, and chronic pain. The efficient pain control leads to the earlier mobilization of patients and their faster recovery, thus leading to shorter hospitalization and lower treatment expenses [2,3,4,5]. Because of these, it is highly necessary to achieve adequate postoperative analgesia, especially during the first 48 h after surgery, with minimum side effects. Pain experienced after an abdominal hysterectomy is categorized as severe pain [6]. It has also been proven that pain treatment in women is a more complex process, and that women have experienced higher pain intensity than men in the acute postoperative period [7].

Even today, with all the advances in pain medicine, one cannot say that there is an ideal mode of analgesia in perioperative medicine. Intravenous opioids and other analgesics are the most frequently used drugs to alleviate postoperative pain in the first 24 h after surgery. The use of opioids for surgery patients is an especially challenging problem requiring clinicians to balance managing acute pain in the postoperative period and minimizing the risks of persistent opioid use after surgery [8]. In the treatment of moderate to severe acute postoperative pain, opioids have been widely used [9]. The analgesic effect is achieved by activating mu-opioid receptors on the neurons that are part of the pain transmission pathway [10]. Opioid side effects are well known, and they include nausea, vomiting, constipation, vertigo, somnolence, and many others [11]. While trying to avoid these side effects, patients are often underdosed with opioids, and pain control is not adequate, which is why it is important to use the proper opioid medication despite providing multimodal analgesia.

Morphine has been the archetypal analgesic for use in moderate and severe pain. It is also the gold standard against which other injected analgesics are tested. Morphine is an agonist at mu- and kappa-opioid receptors. Opioids appear to exert their effects by increasing intracellular calcium concentration, which, in turn, increases potassium conductance and hyperpolarization of excitable cell membranes. The decrease in membrane excitability that results may decrease both pre- and post-synaptic responses. Respiratory depression, nausea and vomiting, hallucinations, and dependence may complicate the use of morphine, especially after intravenous administration [12].

Tapentadol is an analgesic with a double central mechanism of action (mu-opioid receptor agonism and inhibiting noradrenalin uptake). Tapentadol has a direct analgesic effect, without pharmacologically active metabolite [13,14]. The administration of opioid antagonists does not inhibit tapentadol efficacy, which proves the double mechanism of action of this medication [15,16]. In Serbia, it is available in the form of tablets, as immediate release (IR) formulation and extended release (ER) formulation. Studies have shown that tapentadol IR can provide adequate postoperative analgesia for moderate to severe pain, which is similar to analgesia provided by oxycodone IR, with a lower incidence of nausea, vomiting, and constipation [17,18,19].

This study aimed to examine the efficacy of tapentadol IR in the treatment of acute postoperative pain after abdominal hysterectomy in comparison with analgesia provided by intravenously given morphine hydrochloride, as well as to examine the patient’s contentment with analgesia and the frequency of opioid-related side effects in observed patients. 

Not many similar studies have been published up to now. To the best of our knowledge, the published studies on the analgesic effects of tapentadol IR are mainly performed on orthopedic and dental patients [20,21]. We were able to compare our results with two studies carried out on patients after hysterectomy who were treated with tapentadol IR and oxycodone in the postoperative period, where the first one was performed in patients after laparoscopic hysterectomy and the other in patients after open abdominal hysterectomy [22,23]. There are no available studies that compared morphine hydrochloride and tapentadol IR yet to the best of our knowledge.

## 2. Materials and Methods

Having obtained the approval for the research of the Board of Ethics of the Oncology Institute of Vojvodina (4/20/2-3185/2-6), prospective research has been conducted in the Anesthesia and Critical Care Department of the Oncology Institute of Vojvodina, Serbia. All of the patients who consented to participate in this study have given their documented written consent. The study population was formed out of the patients who underwent total abdominal hysterectomy with bilateral salpingo-oophorectomy because of the malignancy, and it included a total number of 100 patients. There were 6 patients excluded from this study for not meeting the inclusion criteria. This prospective observational study was conducted for five months (January–May 2021) until two cohorts with 50 patients in each were formed. 

The two cohorts had different types of postoperative analgesia, tapentadol IR or morphine hydrochloride, and the effects were observed for 24 h postoperatively. Inclusion criteria for this study were written consent to participate in this study, patients who are opioid naive, and patients who had a total abdominal hysterectomy with bilateral salpingo-oophorectomy.

Patients were excluded if they did not consent to participate in this study; they had allergic reactions to analgesics that are part of standard analgesia protocol; they are patients with opioid dependency; they are patients who are under treatment for chronic pain; and they are patients in whom intraoperative findings suggested that the surgery cannot be performed (metastasis and infiltration of other abdominal organs). 

The following data were entered in the individual study protocol: age, sex, weight, preexisting comorbidities, and ASA (American Society of Anesthesiologists) physical status classification. BMI (body mass index) was calculated for both groups. Patients were operated on under general balanced anesthesia. All the women received transabdominal hysterectomy with bilateral salpingo-oophorectomy through a traditional abdominal incision in the lower abdomen, and the surgical procedure was similar in all patients.

All patients were premedicated with midazolam (0.05 mg/kg intravenously). The anesthesia induction was achieved with propofol (1.5–2.5 mg/kg), fentanyl (1.5 μg/kg), and a non-depolarizing muscle relaxant rocuronium (0.6–1.0 mg/kg) for muscle relaxation for intubation, as well as during the surgery at a dose adequate to maintain the relaxation. Anesthesia was maintained with inhalational sevoflurane (1.2–1.4% end-tidal or 1.0–1.2 MAC). The lung ventilation was ensured with the gas mixture O2:N2O 40:60 so that EtCO2 < 38 mmHg. Electrocardiogram, blood pressure, heart frequency, SpO2, EtCO2, and body temperature were monitored during the surgery. Intraoperative analgesia was provided with boluses of fentanyl (50–100 mcg) according to patient needs. Anesthetics were administered in such a way as to provide satisfactory anesthesia, blood pressure values, and heart frequency in the values ±30% as compared with the preoperative values. Drugs for the reversal of neuromuscular blockade (neostigmine 0.02 mg/kg and atropine 0.01 mg/kg) were given at the end of the surgery. A single dose of antiemetic medication ondansetron 4 mg intravenously (i.v.) was given at the end of the surgery to each patient.

After the surgery, patients were admitted to the ICU ward, where they were provided with multimodal analgesia. Upon admission to the ICU, the patients from both groups were given 5 mg of morphine hydrochloride i.v. and, after an hour, 1 g of acetaminophen i.v. for immediate pain control, which is part of a standard analgesia protocol at our institution. Our institution has an analgesia protocol for total abdominal hysterectomy, where analgesia can be achieved with intravenously given opioid analgesic morphine hydrochloride or with oral opioid analgesic tapentadol immediate release (tapentadol IR) in the form of a tablet. 

The type of analgesia (tapentadol IR or morphine hydrochloride) that the patient obtained in the postoperative period was decided by a board-certified anesthesiologist who was assigned to the specific case, and no randomization into one of the following groups was carried out for the study purpose since both medications are equally used in our institution.

Two hours after the surgery, patients in the tapentadol group were given the first oral dose of 50 mg of tapentadol IR. If the pain intensity was still high one hour after the first dose, they were given the second (rescue) dose of 50 mg. After that, they were given 50–100 mg every 6–8 h, alongside intravenous acetaminophen 1 g q8h (max 3 g/24 h) and ketorolac 30 mg q8h (max 90 mg/24 h) or metamizole 2.5 g q12h (max 5 g/24 h). 

The morphine group received 5 mg of morphine hydrochloride intravenously every 6–8 h, with intravenous acetaminophen 1 g q8h (max 3 g/24 h) and ketorolac 30 mg q8h (max 90 mg/24 h) or metamizole 2.5 g q12h (max 5 g/24 h). 

Patients’ vital functions were monitored in the ICU for the first 24 h after surgery, and the pain intensity was assessed 6, 12, 18, and 24 h postoperatively at rest and on exertion (while coughing) by using an 11-point Numerical Pain Rating Scale (NRS; 0 = ‘’no pain’’, 10 = ‘’worst pain imaginable’’). Pain scores were taken at specific times by one of the medical doctors who were a part of the research team. The contentment with the quality of analgesia was also assessed 24 h after the surgery on a scale from 1 to 4 (1—bad analgesia; 2—not that good analgesia; 3—good analgesia; 4—very good analgesia). Opioid-related side effects were noted in the study protocol: PONV—postoperative nausea and vomiting (mild, moderate, intense, severe), sedation, respiratory depression (hypoventilation with a drop in SpO2 <95%), bradycardia, hypotension, and vertigo.

The data were analyzed and processed by IBM SPSS statistics 10.0 software (SPSS Inc., Chicago, IL, USA) and given in tables and figures created in Word and Excel Microsoft Office 2016 packs. The results were presented using standard statistical methods: frequency (f), arithmetic means (x), standard deviation (SD), value intervals (maximum and minimum), and percentages (%). Pearson correlation analysis was used. The patients’ characteristics were compared with the Student’s *t*-test and χ^2^ test. Statistical significance was assumed at *p* < 0.05.

## 3. Results

Both the tapentadol group and the morphine group had 50 patients in each. There were no statistically significant differences in age, BMI, smoking habits, kinetosis, ASA classification, and the duration of surgery between the groups (Table 1).

Table 2 and Figure 1 show the mean values of pain at rest assessed according to NRS during 24 h after the surgery. 

A marginal statistical difference (*p* = 0.07) was found between the groups only at 24 h after the surgery, where the mean pain score at rest was greater in the tapentadol IR group than in the morphine group. 

Table 3 and Figure 2 show the mean values of pain on exertion (coughing) assessed according to NRS during 24 h after the surgery.

Statistical significance was found when assessing pain 24 h after surgery while coughing, where patients in the tapentadol IR group had significantly higher mean pain scores (*p* < 0.01) than patients in the morphine group. A marginal statistical difference (*p* = 0.06) was found when assessing pain 18 h after surgery on exertion when patients from the tapentadol IR group had higher mean pain scores than those from the morphine group. Statistical significance was also found (*p* < 0.01) in assessing pain grade on exertion, where patients from the tapentadol IR group had higher mean pain grades than patients from the morphine group (Table 4). 

The contentment with analgesia quality in both groups is shown in Table 4. It has been found that patients in the tapentadol group were significantly more content with analgesia quality (χ^2^ = 10.79, df = 2, *p* = 0.005). About 96% of patients in the tapentadol group and 86% in the morphine group assessed their analgesia as “3—good” and “4—very good”. No statistical significance was found regarding mean scores of contentment with analgesia between the two groups (Table 5).

Pearson correlation analysis was carried out to assess the relationship between pain control level (NRS) and patients’ contentment with analgesia for both groups, at rest, as well as on exertion. Pearson correlation analysis (Table 6) showed that patients from both groups who experienced less pain (and had lower NRS scores) were more content with analgesia both at rest and on exertion (all correlations are significant and negative, *p* < 0.05). Correlation coefficients showed a stronger negative correlation in the morphine group.

One or more opioid-related side effects were noticed in 34% (*n* = 17) of patients from the tapentadol group and in 46% (*n* = 23) of patients from the morphine group. Figure 3 shows the frequency of opioid-related side effects in both groups. Vertigo appeared significantly more in patients from the morphine group (χ^2^ = 8.7, df = 1, *p* = 0.003). There were no significant differences between the groups for postoperative nausea and vomiting, sedation, respiratory depression, bradycardia, and hypotension.

PONV was experienced in 12% (*n* = 6) patients in the tapentadol group and 16% (*n* = 8) patients in the morphine group. A mild level of nausea was not experienced in any group. A moderate level of nausea was experienced by 6% (*n*= 3) from the tapentadol group and 8% (*n* = 4) from the morphine group. Intense nausea with vomiting was experienced by 6% (*n* = 3) of patients in the tapentadol group and 8% (*n* = 4) in the morphine group. Severe nausea and vomiting were not noticed in any of the groups. 

The average number of tapentadol tablets received in 24 h was 5.22 ± 0.68 (261 ± 33.94 in mg) (Table 5). About 86% of patients (*n* = 43) received more than 4 tablets (200 mg) in 24 h, and they did not have significantly more opioid-related side effects experienced than those who had 4 tablets (χ^2^ = 5.08, df = 1, *p* = 0.02).

The average morphine hydrochloride dose that patients received in 24 h was 20.20 ± 4.60 mg. About 40% (*n* = 20) had more than 20 mg of morphine hydrochloride in 24 h (Table 4), and they had significantly more opioid-related side effects than those patients who had less than 20 mg in 24 h (χ^2^ = 4.84, df = 1, *p*= 0.03). There was no correlation found between experiencing opioid-related side effects and having a smoking habit, suffering from kinetosis, or belonging to a specific age group.

## 4. Discussion

The purpose of this study was to compare the analgesic efficacy, side effects, and patients’ satisfaction with two opioid drugs: the commonly used morphine and the newly introduced tapentadol IR. Due to the fact that tapentadol IR is not available in the intravenous form, our patients started receiving the tablet form two hours after surgery. Both substances had been compared in many studies, but we failed to find any direct comparison of these two drugs after abdominal hysterectomy. We have shown that tapentadol was not significantly different from morphine hydrochloride for the treatment of acute postoperative pain after abdominal hysterectomy, except for 24 h postoperatively when patients in the tapentadol IR group had significantly higher mean pain scores on exertion. Tapentadol was favorable in terms of less vertigo, but there were no significant differences between the groups for postoperative nausea and vomiting, sedation, respiratory depression, bradycardia, and hypotension, although they developed slightly more frequently in the morphine group. 

To reduce the incidence of postoperative complications and improve the postoperative outcome, it is very important to provide adequate analgesia to each patient [3,4]. The efficacy of tapentadol IR has been studied in several clinical trials. Stegman et al. [24] conducted a randomized, double-blind, phase 2 study in orthopedic patients to evaluate the tolerability and efficacy of tapentadol IR. Tapentadol 50 mg resulted in a significant decrease in total pain relief after 24 h on evaluation day 2 (TOTPAR24) in comparison with the placebo. Another study found that mean total pain relief scores over 8 h (TOTPAR8) were significantly higher with placebo than with tapentadol 50 mg, 75 mg, 100 mg, and 200 mg after a single dose in treating moderate to severe dental postoperative pain. Patients who received tapentadol 50 mg experienced a 50% reduction in pain in 46% of cases and those who received morphine in 64.7% of cases [25]. Our results showed that pain control in both tapentadol and morphine groups was good, and that tapentadol IR had similar analgesic effects as morphine. Mean NRS scores decreased during the time in rest and in the exertion in both groups. In the tapentadol IR group, mean pain scores in rest decreased from 2.88 (6 h after surgery) to 0.50 (24 h after surgery), and in the morphine group from 2.82 to 0.22. In exertion, mean NRS scores in the tapentadol IR group decreased from 4.30 (6 h after surgery) to 1.60, and in the morphine group from 3.74 to 0.56, which is similar to published literature [22,23]. In the study of Daniels et al. [17], a reduction in pain intensity of 50% or greater was found in 56.7 to 70.3% of patients receiving tapentadol 50, 75, or 100 mg, and multiple doses of tapentadol significantly reduced acute, postsurgical pain in comparison with the placebo. 

Satisfaction with analgesia may be the most important factor when opting for postoperative analgesia. As much as 96% of patients in the tapentadol IR group assessed the quality of analgesia and their satisfaction with it as “3—good” and “4—very good” versus 86% in the morphine group, and the statistical difference was found between the two groups, where patients were more content with analgesia in the tapentadol IR group. Our results correlate with a similar published study where 61.4% of patients in the tapentadol 50 mg group and 86.4% of patients in the tapentadol 100 mg group rated pain control as good, very good, or excellent [24]. Although it is a misconception that low pain intensity scores are indicative of positive patient satisfaction, and there is a biopsychosocial element to pain that should not be ignored, our study found that patients from both groups who experienced less pain (and had lower NRS scores) were more content with analgesia both at rest and on exertion. When assessing the contentment with analgesia, patients probably considered the negative experience they had with opioid-related side effects. PONV, sedation, respiratory depression, hypotension, and vertigo were more noticed in the morphine group. Discomfort that is caused by opioid-related side effects can affect patients’ assessment of the quality of analgesia. Since they are subjectively reporting it, they may have a feeling of being dissatisfied in the end, if they were nauseous or had some other discomfort, even though their pain grades were low when assessed. The satisfaction with analgesia was only assessed at one time, which is 24 h after surgery. In addition, it highly depends on informing the patients about the procedure and postoperative course before surgery. A qualitative interview study by Mubita WB et al. showed the importance of giving clear and detailed information about pain therapy, pain assessment routines, and the nature of the operation to patients in order to help them control their pain. Although the participants had different views on what they felt was important information on pain management, this indicated that information delivery about pain control should be tailored to individual patient needs [26]. 

Opioid-related side effects were found in 34% of patients in the tapentadol group, and 46% of them in the morphine group (but no statistical significance was found), which is less than that in the published literature, where the incidence of opioid side effects in the tapentadol 50 mg group was as high as 70% [17]. In our study, vertigo was significantly more experienced in the morphine group, which is similar to the results found in the literature where the incidence of vertigo was lower in tapentadol 50 mg in comparison with the oxycodone 10 mg group [24]. Another study showed that 52.3% of patients who were in the tapentadol IR group and 58% of patients in the oxycodone IR group reported at least one treatment-emergent side effect. Nausea and vomiting were experienced by 15.9% and 15.9% of patients in the tapentadol IR group and 20.7% and 24.7% of patients in the oxycodone group, and that is also similar to our findings [27].

A limitation of our study might be a small sample of patients and the fact that it was conducted in one center for only a few months. It is difficult to observe rare side effects in a small study sample without doing the sample size calculation. A similar study with a higher number of subjects would contribute to the strength of our conclusions. Another limitation is that the study population was composed of only female patients, as we evaluated the efficacy of tapentadol IR in gynecologic patients. Moreover, our study findings may be limited by dosing the medication every 6–8 h and constant monitoring during treatment, as these patients spent the first 24 h in the ICU, which does not mirror the actual pain treatment in clinical practice. Pain scores were self-reported by patients, as well. We did not separately assess the patients who were given the rescue dose of tapentadol IR after their pain control was low after the initial dose of 50 mg. Additional limitations include the lack of evaluation of the long-term effects of analgesia up to 48 or 72 h.

## 5. Conclusions

According to our study, it can be concluded that tapentadol IR and morphine hydrochloride are both effective analgesics used in the first 24 h after transabdominal hysterectomy with bilateral salpingo-oophorectomy. Pain control on exertion was significantly better in the morphine group only at one time point, which is 24 h after surgery. Overall satisfaction of patients with analgesia was good. Patients in the tapentadol IR group were significantly more content with analgesia quality. Tapentadol IR resulted in less dizziness than morphine, but there were no significant differences found for PONV, sedation, respiratory depression, bradycardia, and hypotension.

## Figures and Tables

**Figure 1 medicina-59-01800-f001:**
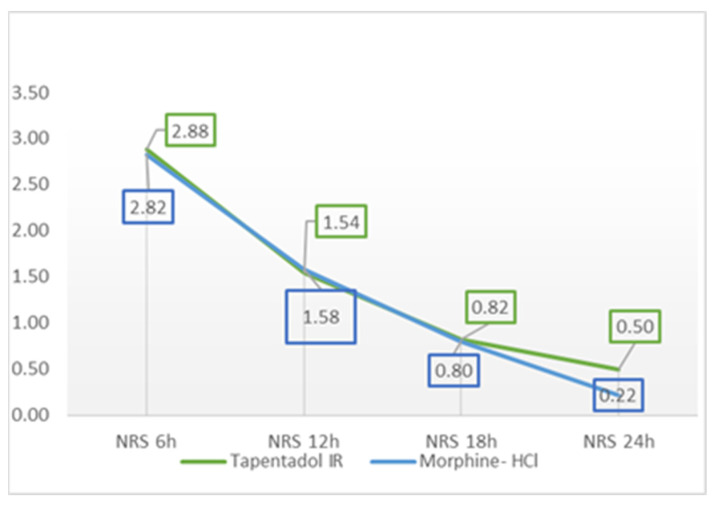
Mean NRS at rest, 6, 12, 18, and 24 h after surgery; NRS—Numeric Rating Scale.

**Figure 2 medicina-59-01800-f002:**
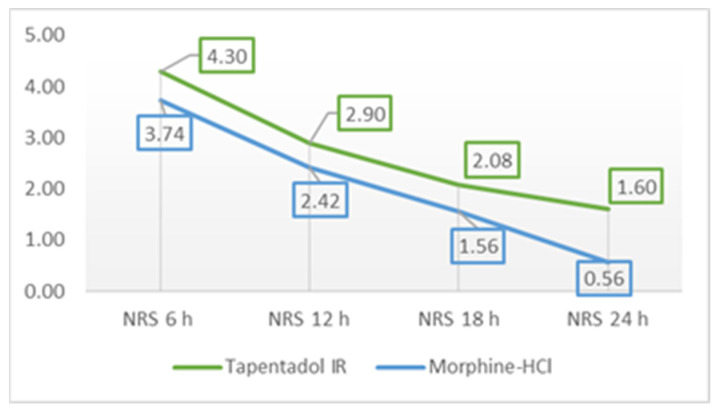
Mean NRS on exertion, 6, 12, 18, and 24 h after surgery; NRS—Numeric Rating Scale.

**Figure 3 medicina-59-01800-f003:**
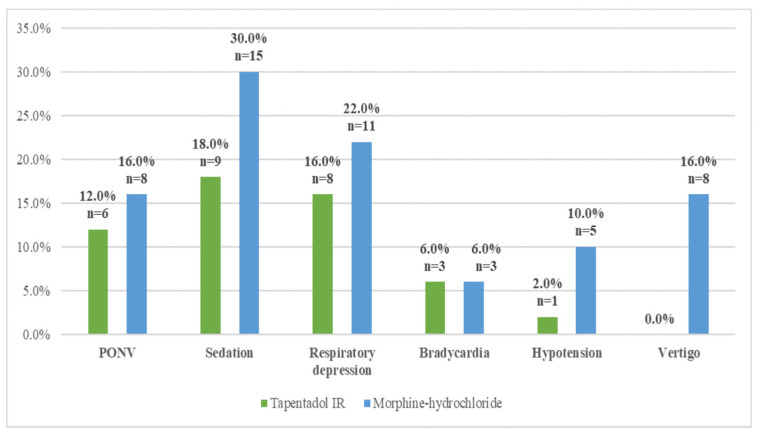
Opioid-related side effects in the tapentadol group and morphine group.

**Table 1 medicina-59-01800-t001:** Patient characteristics.

Parameters	Tapentadol IR (*n* = 50)	Morphine Hydrochloride (*n* = 50)	Test	*p*
Mean age (years ± SD)	58.95 ± 10.98	60.48 ± 9.23	*t* = 0.75	0.45
18–34 years *n* (%)	0 (0)	2 (4)		
35–48 years *n* (%)	12 (24)	7 (14)		
49–64 years *n* (%)	19 (38)	19 (38)		
65–79 years *n* (%)	18 (36)	19 (38)		
>80 years *n* (%)	1 (2)	3 (6)		
BMI (kg/m^2^ ± SD)	27.34 ± 5.98	28.99 ± 7.04	*t* = −1.26	0.21
Smoking—yes *n* (%)	19 (38)	16 (32)	χ^2^ = 0.65	0.53
Kinetosis—yes *n* (%)	2 (4)	0 (0)	χ^2^ = 0.59	0.56
ASA I *n* (%)	4 (8)	1 (2)		
ASA II *n* (%)	37 (74)	35 (70)		
ASA III *n* (%)	9 (18)	14 (28)		
Duration of surgery (min ± SD)	120.00 ± 49.59	138.30 ± 48.69	*t* = 1.92	0.07

**Table 2 medicina-59-01800-t002:** Mean NRS (at rest) 6, 12, 18, and 24 h after surgery (SD—standard deviation; NRS—Numeric Rating Scale).

Parameters	Tapentadol IR				Morphine Hydrochloride				*t*	*p*
	Mean	SD	Min	Max	Mean	SD	Min	Max		
NRS 6 h	2.88	1.94	0.00	7.00	2.82	1.38	0.00	7.00	0.18	0.86
NRS 12 h	1.54	1.54	0.00	6.00	1.58	0.99	0.00	4.00	0.16	0.88
NRS 18 h	0.82	1.18	0.00	4.00	0.80	0.98	0.00	4.00	0.09	0.93
NRS 24 h	0.50	0.97	0.00	4.00	0.22	0.50	0.00	2.00	1.81	0.07

**Table 3 medicina-59-01800-t003:** Mean NRS on exertion (coughing) 6, 12, 18, and 24 h after surgery (SD—standard deviation; NRS—Numeric Rating Scale).

Parameters	Tapentadol IR				Morphine Hydrochloride				*t*	*p*
	Mean	SD	Min	Max	Mean	SD	Min	Max		
NRS 6 h	4.30	2.14	0.00	9.00	3.74	1.32	1.00	8.00	1.58	0.12
NRS 12 h	2.90	1.82	0.00	6.00	2.42	0.92	1.00	5.00	1.67	0.10
NRS 18 h	2.08	1.57	0.00	6.00	1.56	1.07	0.00	5.00	1.94	0.06
NRS 24 h	1.60	1.59	0.00	7.00	0.56	0.70	0.00	2.00	4.23	<0.01

**Table 4 medicina-59-01800-t004:** Contentment with analgesia quality in tapentadol IR and morphine hydrochloride groups.

Parameters		Group			χ^2^	*p*
		Tapentadol IR		Morphine Hydrochloride		
Contentment with analgesia	2—not that good	*n*	2	7		
		%	4	14		
	3—good	*n*	35	19	10.79	0.005
		%	70	38		
	4—very good	*n*	13	24		
		%	26	48		
Total		*n*	50	50		
		%	100	100		

**Table 5 medicina-59-01800-t005:** Additional data.

Parameter	Tapentadol IR	Morphine Hydrochloride	Test	*p*
Opioid-related side effects—yes *n* (%)	17 (34%)	23 (46%)	χ^2^ = 1.50	0.22
Contentment with analgesia	3.22 ± 0.51	3.34 ± 0.71	*t* = −0.90	0.37
Pain grade at rest	1.44 ± 1.11	1.36 ± 0.73	*t* = 0.41	0.69
Pain grade on exertion	2.72 ± 1.43	2.07 ± 0.80	*t* = 2.85	<0.01
Total pain grade	2.08 ± 1.19	1.71 ± 0.75	*t* = 1.82	0.08
Doses of medication	5.22 ± 0.68 (min 4 to max 6 tablets) 261 ± 33.94 (min 200 mg to max 300 mg)	20.20 ± 4.60 (min 10 to max 32.5 mg)		

**Table 6 medicina-59-01800-t006:** Relationship between patients’ contentment with analgesia and pain control (NRS—Numeric Rating Scale).

		Contentment with Analgesia—All (*n* = 100)	Contentment with Analgesia— Morphine Hydrochloride (*n* = 50)	Contentment with Analgesia—Tapentadol IR (*n* = 50)
Contentment with analgesia/NRS	Pearson Correlation	1	1	1
	*p*			
NRS_rest_6h	Pearson Correlation	−0.560	−0.679	−0.511
	*p*	0.000	0.000	0.000
NRS_rest_12h	Pearson Correlation	−0.438	−0.684	−0.286
	*p*	0.000	0.000	0.044
NRS_rest_18h	Pearson Correlation	−0.473	−0.535	−0.441
	*p*	0.000	0.000	0.001
NRS_rest_24h	Pearson Correlation	−0.250	−0.154	−0.351
	*p*	0.012	0.286	0.012
NRS_exertion_6h	Pearson Correlation	−0.476	−0.680	−0.362
	*p*	0.000	0.000	0.010
NRS_exertion_12h	Pearson Correlation	−0.418	−0.679	−0.307
	*p*	0.000	0.000	0.030
NRS_exertion_18h	Pearson Correlation	−0.452	−0.571	−0.380
	*p*	0.000	0.000	0.006
NRS_exertion_24h	Pearson Correlation	−0.345	−0.425	−0.370
	*p*	0.000	0.002	0.008

## Data Availability

The data presented in this study are available on request from the corresponding author. The data are not publicly available due to ethical and privacy considerations.

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
