# Peer review of "Tapentadol Immediate Release (IR) versus Morphine Hydrochloride for Postoperative Analgesia of Patients Undergoing Total Abdominal Hysterectomy—A Prospective Cohort Study"

_medicina, 2023, doi:10.3390/medicina59101800_

Round 1
Reviewer 1 Report
With pleasure, I read the paper titled: “Tapentadol immediate release (IR) versus morphine hydrochloride for postoperative analgesia of patients undergoing total abdominal hysterectomy – a prospective cohort study” by Starčević and colleagues.
Generally speaking, the subject matter is important to a wide array of readers, including those with interest in gynecologic oncology, general gynecology, anesthesiology, and palliative care. The topic deals with a very important postoperative endpoint, which is postsurgical pain relief. The paper is novel with regards to being apparently the first exploration of tapentadol IR compared with morphine HCl I patients with TAH.
I have several issues that should be addressed prior to offering an acceptance.
The results section of the abstract are not reported in a smooth way. You may refer to my points below. Conclusion is not pretty right and should be rephrased.
The methods section needs further details to allow for better transparency in reporting and reproducibility. You may need to clearly indicate various aspects relating to study design regarding randomization and blinding (blinding of patients and blinding of outcome assessors). Why was there no sample size calculation?
The results section also requires major changes regarding the reporting of finding as the present format is not acceptable. Authors are required to summarize the sociodemographic of patients per group and conduct statistical tests to pinpoint if such differences exist at the baseline level. The endpoints such as operative time, pain score, and contentment with analgesia and others should be reported using standard statistical approach using mean ± SD or median (IQR) and ranges—this MUST be done. There lots of reprition between results and discussion, and this should be rectified during the revised draft.
The conclusion is not pretty right and should be rephrased. When I examine Figures 1/2 I get a sense that morphine HCl is better than tapentadol IR at all time points tested. Isn’t it?
Author Response
We are grateful to the reviewers for their insightful comments on our paper. We have been able to incorporate changes to reflect most of the suggestions provided by the reviewers. Here is our point-by-point response to reviewers comments and suggestions:
-The Method section was updated with more data and details for better transparency and reproducibility.
-We did not do the sample size calculation. We are highly aware of its importance in getting relevant statistical results. We observed these patients until we reached the number of 50 in each group (6 of them were excluded for not meeting the criteria). Both medications are used equally at our Institution and are a part of standard analgesia protocol for abdominal hysterectomy. The board-certified anesthesiologist who was assigned to the specific case decided the type of postoperative analgesia the patient will get, between these two options. We do not provide routinely regional anesthesia for postoperative pain management for hysterectomy patients.
-The Results section was rewritten and improved, additional statistics were done.
-We presented the demographic characteristics of patients in a table, and have done the analysis to see if there were significant differences between the two groups (table 1).
-We improved our statistical analysis of data by presenting the mean pain scores at specific time intervals, with calculated standard deviation and performed the t-test to see if there were significant differences between the groups. The results were displayed as tables 2 and 3, and figures 1 and 2.
-Table 4 that represents additional data is also added to the manuscript, and it contains more statistical analysis of our data.
-Figure 3 which was part of the previous version of manuscript is unchanged.
-Table 5 was added and it contains the results of Pearson Correlation analysis of pain grades and analgesia contentment.
-For the contentment with analgesia, we added scores 1-4 for describing the contentment.
-Fig.4 which shows the frequency of opioid-related side effects was included in the previous version of the manuscript. Statistical analysis was done to compare the two groups in terms of opioid-related side effects, but statistical significance was found only for vertigo in morphine group.
-The Discussion section was rewritten, and it included more references and comparison of our results. The first paragraph was revised and it highlights the purpose and summarizes the principal findings of the study. Additional limitations to the study were acknowledged.
-The conclusion is that analgesia was better on exertion in morphine group, but only at one time (24 hours after surgery). There was no statistical difference found in all the other times when the pain was assessed. This section has been rewritten, as suggested.
Kind regards, and thank you so much for your time!
Sanja Starčević
Reviewer 2 Report
Starčević et al examined the analgesic efficacy of tapentadol immediate release versus morphine for management of pain of patients who received total abdominal hysterectomy. The topic is clinically relevant to the journal of “Medicina” and in line with the scope of the Special Issue. A key novelty of article is being the first presumably to directly compare tapentadol versus morphine in the context of hysterectomy. Overall, the manuscript reads well, however, some changes are must to improve the manuscript regarding methods and presentation of findings as outlined below. After consideration of these major changes, the manuscript can be reconsidered for publication.
Abstract. Please spell out all abbreivations, such as IR and NRS. The results section should be re-written as it is confusing regarding the terms higher grades (5 and 6) and lower grades (2 and 3). I recommend using the average plus/minus standard deviations for the numerical pain score and level of contentment with analgesia. You can, if wish, to include proportion of patients who had pain scores less than for example 5 in each group but keep it simple.
Introduction. The authors need to highlight the significance of their work. Is this the first-ever study on tapentadol versus morphine in patients undergoing abdominal hysterectomy. Also, I would recommend highlighting the specific conditions or procedures whereby tapentadol has been previously used. Lastly, I encourage concluding the introduction section with some proposed hypotheses.
Methods. Had all patients undergone only total abdominal hysterectomy with no other additional procedures such as salpingo-oophorectomy (unilateral/bilateral), etc? Were the patients randomized into two groups? Who did survey the patients regarding the pain scores?
Results. The baseline characteristics of patients should be presented in a table format, as this will make it easier for readers to follow. You may need to quantify if the differences in mean age, length of operation, and others were statistically significant. Figures 2 and 3 are good, however, the authors may need to report the finding more in an objective fashion by specifically dissecting down the number and proportion of each NRS score in a table format and always provide the average plus/minus standard deviation value at the different time points. When were patients asked to express their contentment with analgesia quality; is it at 24 hours? For data regarding the grade of nauseas/vomiting, please provide the numbers and percentages consistently. How did the authors define respiratory depression as a side effect as the percentage was substantially high? The authors need to better report the number of doses received using average plus/minus standard deviation and ranges (minimum to maximum) too.
Discussion. The first paragraph should be revised to briefly highlight the purpose and summarize the principal findings of the study. Also, do not repeat the specific numbers/percentages that are found in the results again in the discussion as this is unnecessary reportion and should be avoided. From Figures 2 and 3, it looks like morhohne group had better analgesic effects, however, the extent of contentment with analgesia was better in tapentadol, which does not seem quietly to make sense—more explanation is needed regarding this aspect. Additional limitations that need to be acknowledged is the lack to evaluate the long-term effects of analgesia up to 48 or 72 hours. Additional limitations include the retrospective study design and the self-reported pain score by patients.
Conclusion. The conclusion does not really match the presented results and I encourage revising it. The second sentence is not right and should be avoided.
Minor English editing is required
Author Response
We are grateful to the reviewers for their insightful comments on our paper. We have been able to incorporate changes to reflect most of the suggestions provided by the reviewers. Here is our point-by-point response to reviewers comments and suggestions:
-The Abstract section was improved as suggested, with all the abbreviations spelled out. The results section was re-written since the main study results were also presented in a different way.
-We highlighted the significance of our work in the Introduction section, and highlighted the procedures where tapentadol has been previously used and studied.
-The Method section was updated, with more details added for better transparency and reproducibility. We explained that we observed patients who were given morphine or tapentadol, since both medications are used equally in our Institution. We did not specificaly randomize patients in one of the two groups. The board-certified anesthesiologist who was assigned to a specific case was the one who decided on the type of postoperative analgesia the patient will receive. So, we observed these patients equally and collected the data from them. Also, numerical Pain Rating Scale is used for all the patients in our ICU on a daily basis. We ask patients about their pain grades routinely during the stay. Our team surveyed the patients included in the study. All the women had transabdominal hysterectomy with bilateral salpingo-oophorectomy through a traditional abdominal incision in the lower abdomen.
-The Results section was rewritten and improved, additional statistics were done.
-We presented the demographic characteristics of patients in a table as suggested, and have done the analysis to see if there were significant differences between the two groups (table 1).
-We improved our statistical analysis of data by presenting the mean pain scores at specific time intervals, with calculated standard deviation and performed the t-test to see if there were significant differences between the groups. The results were displayed as tables 2 and 3, and figures 1 and 2.
-Patients were asked to express their contentment with analgesia quality at 24 hours after the surgery.
-Respiratory depression was defined as hypoventilation with a drop in SpO2 <95% and it is added to the Methods section.
-Table 4 that represents additional data is also added to the manuscript, and it contains more statistical analysis of our data.
-Figure 3 which was part of the previous version of manuscript is unchanged.
-Table 5 was added and it contains the results of Pearson Correlation analysis of pain grades and analgesia contentment.
-For the contentment with analgesia, we added scores 1-4 for describing the contentment.
-Fig.4 which shows the frequency of opioid-related side effects was included in the previous version of the manuscript. Statistical analysis was done to compare the two groups in terms of opioid-related side effects, but statistical significance was found only for vertigo in morphine group.
-The number of doses was reported as suggested, by using average plus/minus SD and ranges.
-The discussion section was improved as suggested. The first paragraph briefly highlights the purpose and summarizes the principal findings of the study. It has been explained why the contentment with analgesia was better in the tapentadol group. But also, when different statistical analysis was used, it was showed that morphine group was slightly more content. We reported both findings. If sample size was bigger, we could maybe get some better data.
-Additional limitations to the study were acknowledged as suggested.
-We did not perform the sample size calculation for the study.
-The conclusion is that analgesia was better on exertion in morphine group, but only at one time (24 hours after surgery). There was no statistical difference found in all the other times when the pain was assessed. This section has been rewritten.
Kind regards, and thank you for your time!
Reviewer 3 Report
The paper is interesting to gynecologic oncology and anesthesiologist physicians. It is a good paper. However, it needs some improvements in reporting the results in tables and figures and texts with better statistics. Although 100 patients is a good number, however, the authors should mention if sample size was calculated or not. The conclusion should be that morphine hydrochloride was better than tapentadol immediate realse based on figure 2 and figure 3. The data on contentment with analgesia quality should be reported using a continuous variable (like score from 1-4) as in its current standing is difficult to inform which method was better. Statistics should be done to tell if the adverse events were statistically different between both groups.
Author Response
We are grateful to the reviewers for their insightful comments on our paper. We have been able to incorporate changes to reflect most of the suggestions provided by the reviewers. Here is our point-by-point response to reviewers comments and suggestions:
-The Results section was rewritten and improved, additional statistics were done.
-We presented the demographic characteristics of patients in a table, and have done the analysis to see if there were significant differences between the two groups (table 1).
-We improved our statistical analysis of data by presenting the mean pain scores at specific time intervals, with calculated standard deviation and performed the t-test to see if there were significant differences between the groups. The results were displayed as tables 2 and 3, and figures 1 and 2.
-Table 4 that represents additional data is also added to the manuscript, and it contains more statistical analysis of our data.
-Figure 3 which was part of the previous version of manuscript is unchanged.
-Table 5 was added and it contains the results of Pearson Correlation analysis of pain grades and analgesia contentment.
-Fig.4 which shows the frequency of opioid-related side effects was included in the previous version of the manuscript. Statistical analysis was done to compare the two groups in terms of opioid-related side effects, but statistical significance was found only for vertigo in morphine group.
-We did not perform the sample size calculation for the study.
-The conclusion is that analgesia was better on exertion in morphine group, but only at one time (24 hours after surgery). There was no statistical difference found in all the other times when the pain was assessed. The whole Conclusion section was rewritten.
-For the contentment with analgesia, we added scores 1-4 for describing the contentment.
Kind regards and thank you for your time!
Round 2
Reviewer 1 Report
I thank the authors for attending to the suggestions recommended by the reviewers. The authors of the manuscript substantially improved the quality of the report and specifically enhanced the presentation of the methods and results sections. The newly added tables are robust and conveyed the results in a better way. Additionally, the English language appeared to be adjusted.
I am a bit confused why the authors analyzed the satisfaction level of patients by applying the person’s correlation test and it does not make sense to me. The data should be deleted or the analysis should be revisited by performing the common parametric ‘t’ test for numerical values. Because of this matter, I will issue a revision decision. Also, p values are considered significant only when it is below 0.05 (please check methods section and revise your abstract/results/discussion/conclusion sections accordingly). Lastly, the asterisk (*) symbols in tables should be removed irrespective of the level of significance. Apart from these comments, the authors should be congratulated for a well-executed research investigation. I look forward to the revised draft.
Author Response
Dear reviewer,
thank you for a very fast response and very detailed analysis of our work. Here we explain our corrections in the manuscript (which are also highlighted in the attached version of the manuscript):
- We have corrected the mistake we made regarding the statistical difference and p-values. The statistical difference is now assumed at p<0.05.
- The results of chi-square test of analgesia contentment show that p<0.005 and that there is statistical difference between the groups (p is less than 0.05). Also, in order to present the results more clearly, we added a Table 4 which represents the results of a contentment with analgesia for both groups and the chi-square test's results (instead as a Figure).
- We made a typing error in abstract where we stated the p<0.05 instead p<0.005, which we corrected.
- We may have badly presented the results of a Pearson correlation analysis in Table 5. Our aim was to show relationship between patients’ contentment with analgesia and pain control (NRS), not the contentment with analgesia itself. This analysis showed us that patients from both groups who experienced less pain (and had lower NRS scores) were more content with analgesia both at rest and on exertion, with all correlations being significant and negative).
- We have done the t-test for the contentment with analgesia and the results are presented in Table 5.
- The asterisk (*) symbols in tables are removed, as suggested.
Looking forward to hearing from you.
Kind regards,
Sanja Starčević
Reviewer 2 Report
The authors adequately revised their manuscript and carefully addressed the comments from the three reviewers. The manuscript is very solid now. The manuscript can be accepted after addressing two minor comments:
1. Typically speaking, the norm for establishing statistical significance is p-value “less than” 0.05. In the present paper, the authors defined statistical significance based on p-value “equal to” 0.05. Authors should correct that and make sure that statistical significance is assumed at alpha-error or p-value <0.05, and rectify the following statement in the abstract and elsewhere “The subjective feeling of satisfaction with postoperative analgesia was statistically significant in the tapentadol group (p=0.05)”.
2. Table 5 is not analyzed correctly and should be done using student’s t-test considering the variables are numerical (a score of 1 to 4). The use of Person’s correlation is not right.
Very minor English editing may be needed.
Author Response
Dear reviewer,
thank you for a very fast response and very detailed analysis of our work. Here we explain our corrections in the manuscript (which are also highlighted in the attached version of the manuscript):
- We have corrected the mistake we made regarding the statistical difference and p-values. The statistical difference is now assumed at p<0.05.
- The results of chi-square test of analgesia contentment show that p<0.005 and that there is statistical difference between the groups (p is less than 0.05). Also, in order to present the results more clearly, we added a Table 4 which represents the results of a contentment with analgesia for both groups and the chi-square test's results (instead as a Figure).
- We made a typing error in abstract where we stated the p<0.05 instead p<0.005, which we corrected.
- We may have badly presented the results of a Pearson correlation analysis in Table 5. Our aim was to show relationship between patients’ contentment with analgesia and pain control (NRS), not the contentment with analgesia itself. This analysis showed us that patients from both groups who experienced less pain (and had lower NRS scores) were more content with analgesia both at rest and on exertion, with all correlations being significant and negative).
- We have done the t-test for the contentment with analgesia ant the results are presented in Table 5.
Looking forward to hearing from you.
Kind regards,
Sanja Starčević